# A frontal cortex event-related potential driven by the basal forebrain

**David P Nguyen[†], Shih-Chieh Lin***

Neural Circuits and Cognition Unit, Laboratory of Behavioral Neuroscience, National Institute on Aging, National Institutes of Health, Baltimore, United States

**Abstract** Event-related potentials (ERPs) are widely used in both healthy and neuropsychiatric conditions as physiological indices of cognitive functions. Contrary to the common belief that cognitive ERPs are generated by local activity within the cerebral cortex, here we show that an attention-related ERP in the frontal cortex is correlated with, and likely generated by, subcortical inputs from the basal forebrain (BF). In rats performing an auditory oddball task, both the amplitude and timing of the frontal ERP were coupled with BF neuronal activity in single trials. The local field potentials (LFPs) associated with the frontal ERP, concentrated in deep cortical layers corresponding to the zone of BF input, were similarly coupled with BF activity and consistently triggered by BF electrical stimulation within 5–10 msec. These results highlight the important and previously unrecognized role of long-range subcortical inputs from the BF in the generation of cognitive ERPs.

*For correspondence: shih-chieh.lin@nih.gov

Present address: [†]Department of Pharmacokinetics, Pharmacodynamics and Metabolism, Worldwide Research and Development, Pfizer Inc., Cambridge, United States

Competing interests: The authors declare that no competing interests exist.

## Introduction

Event-related potential (ERP) represents the stereotypical response of the electroencephalogram (EEG) activity to an internal or external stimulus, and reflects reproducible and highly coherent large-scale activity patterns in the underlying cerebral cortical network (*Davis, 1939*; *Rohrbaugh et al., 1990*; *Luck, 2005a*; *Luck and Kappenman, 2012*). ERPs have been widely used in both healthy and neuro-psychiatric conditions as robust physiological indices of cognitive functions because the amplitudes of late ERP components are modulated by various cognitive functions including attention (*Hillyard and Kutas, 1983*; *Näätänen, 1988*; *Schreiber et al., 1992*; *Iragui et al., 1993*; *Polich, 1997*; *Herrmann and Knight, 2001*; *Barry et al., 2003*; *Caravaglios et al., 2008*; *Luck and Kappenman, 2012*).

Despite the broad applications of ERP in both basic and clinical research, there remains considerable debate regarding the source mechanisms that generate cognitive ERPs (*Snyder, 1991*; *Miltner et al., 1994*; *Pascual-Marqui et al., 2002*; *Luck, 2005b*; *Nunez and Srinivasan, 2006*; *Cohen et al., 2009*; *Riera et al., 2012*). It is commonly assumed that cognitive ERPs are generated by, and therefore reflect the functions of, local activity within the cerebral cortex. However, inferring the underlying sources based on the skull surface EEG pattern, generally referred to as the 'inverse problem', is difficult and lacks a unique mathematical solution because the same EEG pattern can be generated by many different configurations of underlying sources (*Helmholtz, 1853*; *Luck, 2005b*; *Nunez and Srinivasan, 2006*). For this reason, it has remained difficult to experimentally demonstrate how cognitive ERPs are generated.

While most studies have focused on identifying cortical activities responsible for generating cognitive ERPs, one neglected possibility is that subcortical inputs play a significant role. In this study, we explored an alternative hypothesis that ERPs are driven by subcortical inputs from the basal forebrain (BF), one of the major cortically-projecting neuromodulatory systems (*Semba, 2000*; *Zaborszky, 2002*; *Jones, 2003*). This hypothesis is motivated by recent findings that a subset of BF neurons is robustly activated by motivationally salient stimuli that attract animals' attention (*Richardson and Delong, 1991*; *Lin and Nicolelis, 2008*; *Avila and Lin, 2014*), and that the phasic bursting response

**eLife digest** The vertebrate nervous system coordinates an animal's involuntary and voluntary actions, and is responsible for transmitting signals between different parts of the body. Two different cell types, glial cells and neurons, make up the nervous system: glial cells play a metabolic or structural role, whereas neurons are responsible for physically transmitting the signals. Signals are sent quickly and precisely between neurons in the form of electrochemical waves, and it is this electrical activity that allows researchers to measure and study the communication of signals throughout the body.

At the center of the nervous system is the brain. The brain receives and integrates information from all parts of the body, and coordinates appropriate responses to various stimuli. The outermost layer of the brain, which is known as the cerebral cortex, enables an organism to perceive and interact with the world in meaningful ways, and is also responsible for the body's movement, as well as for thought and cognition. Composed of an estimated 30 billion neurons, the cerebral cortex generates a tremendous amount of electrical activity during sensory, motor or cognitive events.

Researchers interested in evaluating brain function or assessing the level of activity in the cerebral cortex often use a non-invasive technique called electroencephalography. This technique has provided insight into the regions of the brain that process sensory and motor events, but it has been difficult to work out where cognitive events are processed. To date, most studies have tried to identify the region of the cerebral cortex that is responsible for generating the electrical activity associated with cognitive events, ignoring the possibility that other regions of the brain could play a significant role in producing this activity that is observed in the cerebral cortex.

Now Nguyen and Lin have shown that the basal forebrain, a region of the brain located beneath the cerebral cortex, is responsible for generating some of the electrical activity that happens during cognitive events. The experiments involved making measurements on rats as they performed a cognitive task. Nguyen and Lin found that a form of electrical activity called an event-related potential occurred in the frontal lobe of the cerebral cortex at the same time as the activity of single neurons in the basal forebrain. Stimulating neurons in the basal forebrain also resulted in an event-related potential being measured within the frontal cortex. These findings raise the possibility that the impairment of these basal forebrain neurons is involved in conditions such as Schizophrenia, ADHD and Alzheimer's disease, and also in normal cognitive aging.

of these BF neurons is tightly coupled with LFP activity in the frontal cortex (*Lin et al., 2006*). These discoveries led us to the hypothesis that ERPs elicited by motivationally salient stimuli are generated by the bursting response of BF neurons.

To test this hypothesis, we trained rats to perform an auditory oddball task that is commonly used in human ERP studies and observed a robust ERP response in the frontal cortex that was coupled with behavioral performance. We simultaneously recorded skull surface EEG and BF single neuron activity, and investigated whether frontal ERP and BF bursting activity were correlated, in single trials, in terms of amplitude and timing. To better understand how the frontal ERP was generated, we recorded LFPs throughout all layers of frontal cortical regions, and identified the layer-specific LFP response that was coupled with the frontal ERP and BF bursting activity. Finally, we tested whether activating BF via electrical stimulation was sufficient to trigger the layer-specific LFP response in the frontal cortex. The results support that the frontal ERP is correlated with, and likely generated by, subcortical inputs from the basal forebrain.

## Results

We first developed a rat version of the auditory oddball task commonly used in human ERP studies, with the goal of reproducing characteristic ERP responses. In the auditory oddball task (*Figure 1A*), a standard tone (10 kHz) was presented once every 2 s, and occasionally once every 6–14 s a deviant oddball tone (6 kHz) was presented instead. Adult Long Evans rats (n = 8) were trained to discriminate the standard tone from the oddball tone that signaled the availability of liquid reward. After training, rats showed high hit rates toward the oddball tone and a low false alarm rate toward the standard tone (*Figure 1B*).

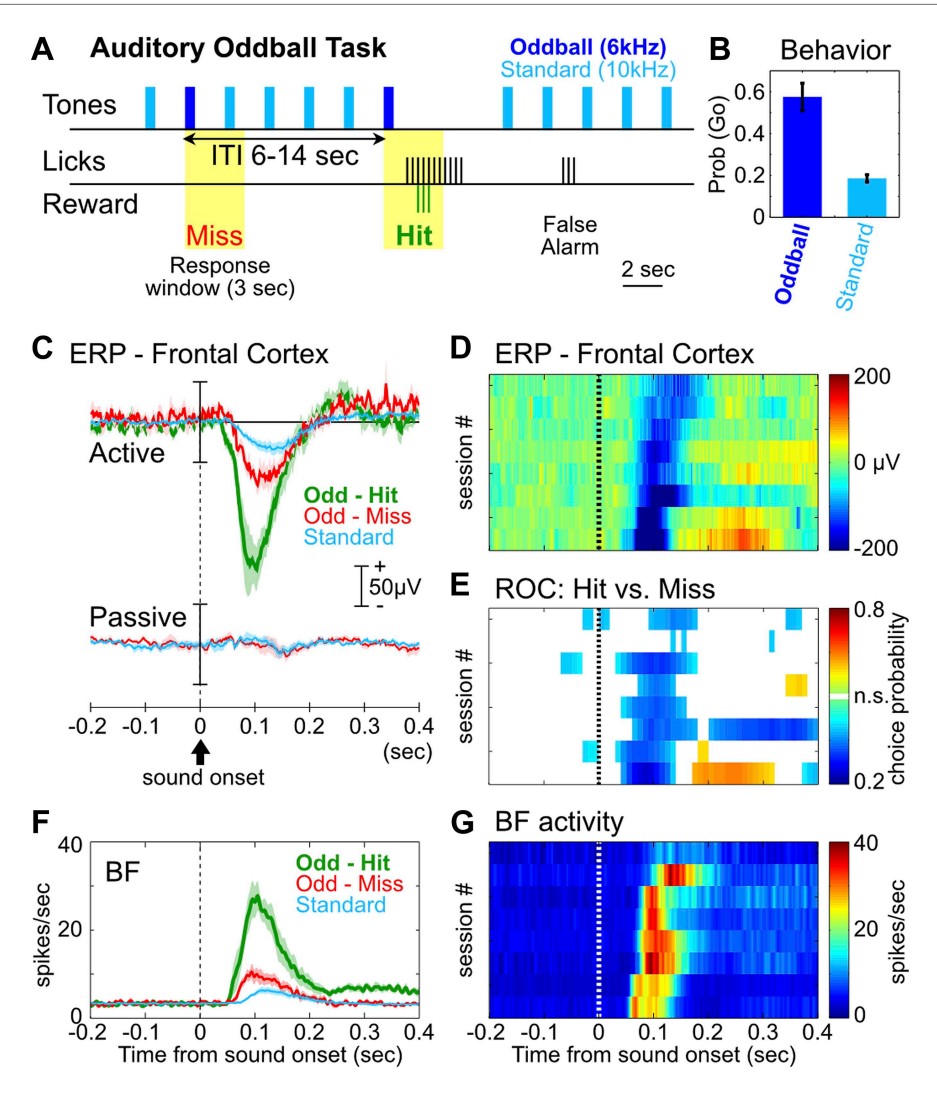

**Figure 1**. The frontal ERP in an auditory oddball task is behaviorally relevant and functionally coupled with BF bursting. (**A**) In the auditory oddball task, a standard tone (10 kHz) was presented once every 2 s, and occasionally once every 6–14 s a deviant oddball tone (6 kHz) was presented that signaled reward if responded to within a 3-s window (yellow). (**B**) Behavioral performance in the oddball task from eight rats (mean ± SEM). (**C**) Grand-average ERPs in the frontal cortex relative to tone onsets in three trial types, plotted separately while rats actively performed the oddball task or when the tones were presented passively without any reward. (**D**) The frontal ERP in hit trials from eight individual sessions (six rats). Sessions were sorted by peak ERP latency, the same as in (**E**) and (**G**). (**E**) Frontal ERP amplitude reliably discriminated hit from miss responses to the oddball tone in individual sessions, calculated based on signal detection theory. Only bins reaching statistical significance (p<0.01) were plotted. (**F**) Grand-average of BF bursting response relative to tone onsets in three trial types. BF bursting occurred at the same time window as the frontal ERP, and showed similar amplitude modulation between trial types. (**G**) Population BF bursting response in hit trials from the same eight sessions.

The following figure supplements are available for figure 1:

**Figure supplement 1**. ERP in the frontal cortex, but not the primary visual cortex, provides information about behavioral performance.

**Figure supplement 2**. BF electrode locations and the classification of BF bursting neurons.

We found a prominent negative ERP response in the frontal cortex when rats correctly responded to the oddball tone (hit) (*Figure 1C–D*). This ERP response started at 50 msec, peaked around 100 msec, and was absent when the same stimuli were not behaviorally relevant (*Figure 1C*). Furthermore, the amplitude of this frontal ERP reliably discriminated hit from miss responses to oddball tones (*Figure 1E*), and also discriminated oddball from standard tones (*Figure 1—figure supplement 1*). These results demonstrate that this frontal ERP is highly relevant for behavioral performance and does not simply reflect sensory processing.

In 120 BF neurons recorded simultaneously with frontal EEG activity, 55% (66/120) showed robust bursting response to the oddball tone in hit trials and were classified as BF bursting neurons (*Figure 1—figure supplement 2*; *Table 1*). These BF neurons showed stronger bursting responses in hit trials than in miss trials, and in oddball trials than in standard trials (*Figure 1F*), consistent with the encoding of motivational salience as previously reported (*Lin and Nicolelis, 2008*; *Avila and Lin, 2014*). Interestingly, the timing of the BF bursting response as well as the modulation of BF bursting amplitude between different trial types (*Figure 1F,G*) were highly similar to those of the frontal ERP, suggesting that BF bursting may be functionally coupled with the frontal ERP.

To determine whether the amplitudes of BF bursting and the frontal ERP were coupled in single trials, we first visualized their relationship by sorting oddball and standard trials based on the amplitude of BF population bursting response in the 50–200 msec window. The example depicted in *Figure 2A* illustrates the frequent observation that trials with strong BF bursting responses were accompanied by strong frontal ERPs, while trials with weak BF bursting were accompanied by weak frontal ERPs. Indeed, significant single trial amplitude correlation (p<0.001) was observed in 95% (63/66) of BF bursting neurons (*Figure 2B*). Furthermore, the amplitude coupling relationships between BF bursting and the frontal ERP were well described by a highly homogeneous linear scaling function

**Table 1.** List of animals and sessions used for each analysis

| Animal ID | BF neurons (Bursting/All) | EEG (Frontal/V1) | LFP | BF stim (awake) | BF stim (isofurane) |
|---|---|---|---|---|---|
| Rat #1 | 7/8 | Frontal/V1 | YES | YES | YES |
| Rat #2 | 10/18 | Frontal/V1 | YES | YES | YES |
| Rat #2 | 8/14 | Frontal/V1 | YES | | YES |
| Rat #3 | 9/14 | – | YES | YES | YES |
| Rat #3 | 5/7 | – | YES | | YES |
| Rat #4 | 4/11 | Frontal/V1 | YES | YES | YES |
| Rat #5 | 16/27 | – | YES | YES | – |
| Rat #6 | 14/31 | Frontal/V1 | YES | – | – |
| Rat #7 | 4/6 | Frontal/V1 | YES | – | – |
| Rat #8 | 6/15 | Frontal/- | – | – | – |
| Rat #8 | 13/17 | Frontal/V1 | | – | – |
| 8 rats 11 sessions | n = 96/168 neurons | FX = 8/V1 = 7 sessions; n = 66/120 | 9 sessions n = 77/136 | 5 sessions | 4 sessions |
| *Figure 1B* | *Figure 1—figure supplement 2* | *Figures 1–3*, *Figure 1—figure supplement 1*, *Figure 3—figure supplement 1* | *Figure 4*, *Figure 4—figure supplement 1*, *Figure 4—figure supplement 2* | *Figure 5*, *Figure 4—figure supplement 1*, *Figure 4—figure supplement 2* | *Figure 5*, *Figure 4—figure supplement 1*, *Figure 4—figure supplement 2* |

A detailed list of all 11 sessions from 8 rats used for the current study. Animal IDs correspond to those used in *Figures 4, 5*, *Figure 4—figure supplement 1*, *Figure 4—figure supplement 2*. Two sessions were recorded in three animals (Rat #2, 3, 8) with BF electrodes positioned at two different depths to record from independent BF neuronal ensembles. The number of neurons indicated by n. Data used for each analysis are indicated in the bottom row. Five rats (Rats #1, 2, 4, 7, 8) were also recorded in one session of passive oddball task in which the stimuli were not behaviorally relevant (*Figure 1C*).

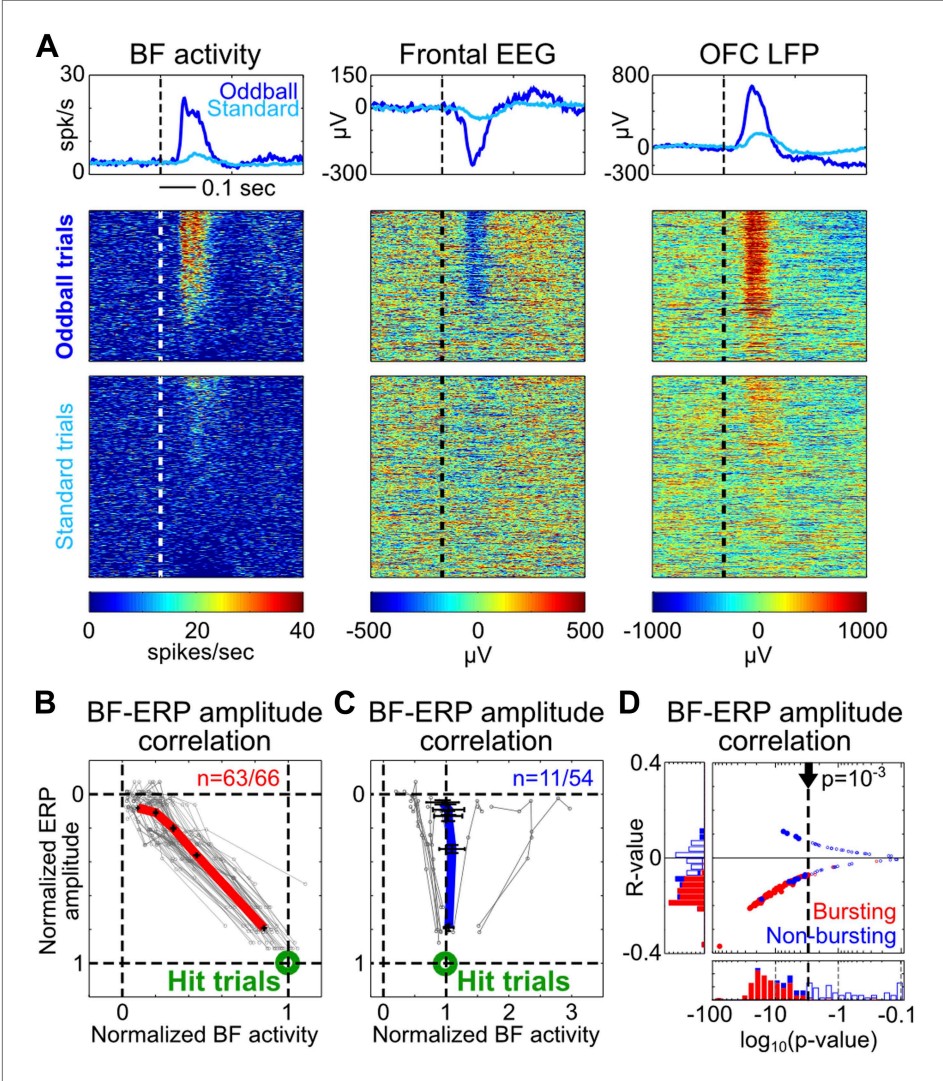

**Figure 2**. Single trial amplitude coupling between BF bursting and frontal ERP. (**A**) An example session showing single trial activities relative to tone onsets in the BF bursting neuron population (left), frontal EEG (middle) and one LFP channel in the deep layer of the frontal cortex (right). Trials were sorted based on the amplitude of BF population bursting response (left). (**B** and **C**) For each BF bursting neuron (**B**) and non-bursting neuron (**C**), the mean amplitudes of BF activity and ERP in each quintile of trials were normalized by their respective average amplitudes in hit trials. Each neuron is represented by a gray line, with the population average (±SEM) for BF bursting neurons and non-bursting neurons shown in red (**B**) or blue (**C**), respectively. 63/66 BF bursting neurons and 11/54 non-bursting BF neurons showed significant single trial amplitude correlation (p<0.001). BF bursting neurons showed a highly homogeneous linear scaling relationship with ERP amplitude. (**D**) The scatter plot and histograms of R- and p-values for BF-ERP amplitude correlations for all BF neurons. Significantly correlated neurons were shown in filled symbols.

(*Figure 2B*) despite the variable bursting amplitude among BF bursting neurons (*Figure 1—figure supplement 2*). This linear scaling relationship indicates that the amplitude coupling between BF bursting and the frontal ERP remains invariant in the face of different types of stimulus (oddball vs standard) or behavioral response (hit vs miss), and therefore likely reflects a true functional coupling between BF bursting and the frontal ERP.

In contrast to the homogeneous amplitude coupling relationship in BF bursting neurons, only 20% (11/54) of non-bursting BF neurons showed significant single trial amplitude correlation with the frontal ERP, which consisted of heterogeneous coupling relationships (*Figure 2C*) that were less correlated

with the frontal ERP compared to BF bursting neurons (*Figure 2D*). This finding is evidence that the frontal ERP is coupled specifically with the subset of BF neurons that encode motivational salience using the phasic bursting response (*Lin and Nicolelis, 2008*; *Avila and Lin, 2014*).

In addition to amplitude coupling, BF bursting and the frontal ERP were also coupled temporally. At the gross temporal scale, BF bursting and the frontal ERP always occurred in the same time window despite the variable latency of the frontal ERP responses between sessions and across trial types (*Figure 1D–G*). These coupling relationships are compatible with two concurrently plausible explanations: First, it may be that both the BF bursting and frontal ERP are controlled by common inputs from un-observed third brain region(s). A second possibility is that the frontal ERP is driven by BF activity with a very short delay.

To further examine these two possibilities, we determined the time lag that gave the maximal cross correlation between each BF bursting neuron and the frontal ERP (*Figure 3*, *Figure 3—figure supplement 1*). This analysis showed that, while the activity of many BF bursting neurons either preceded or

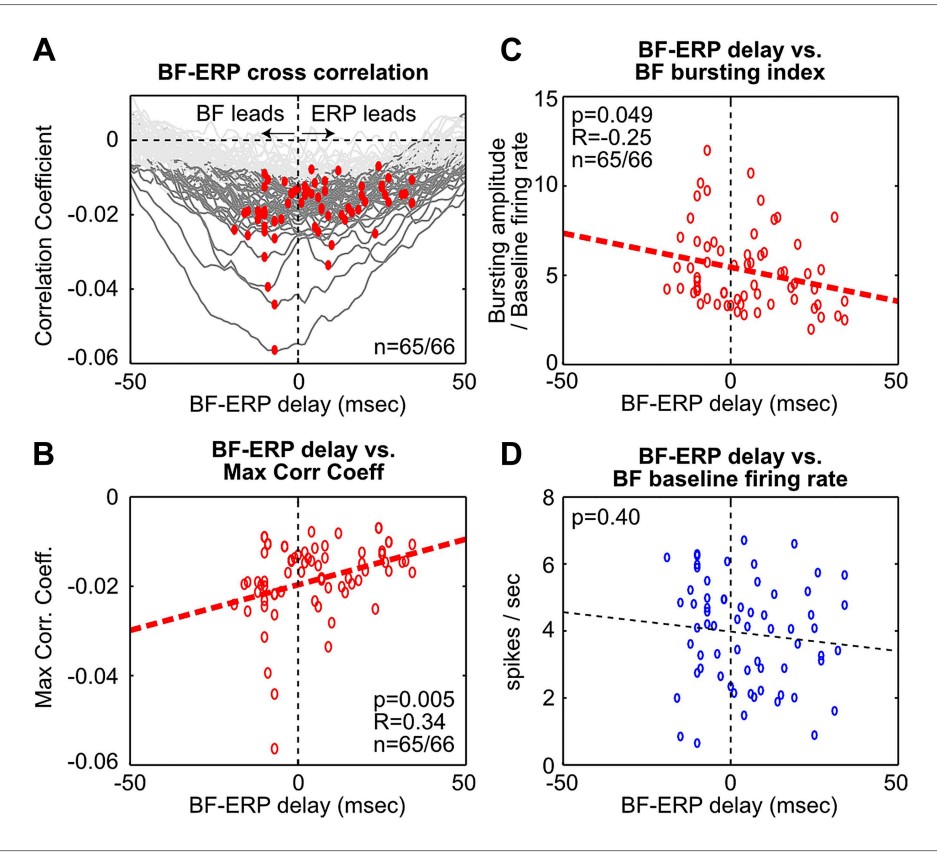

**Figure 3**. Fine temporal relationship between BF bursting and the frontal ERP. (**A**) Cross correlation between the activity of individual BF bursting neurons and the frontal EEG activity. Each BF bursting neuron is represented by one gray line, with dark gray indicating correlation exceeding statistical significance (p<0.01 in 65/66 BF bursting neurons, permutation test) and red dots indicating the maximum correlation. (**B**) Correlation between the BF-ERP delay that produced the maximum correlation against the maximum correlation coefficient in each BF bursting neuron. The significant positive correlation shows that the BF bursting neurons whose activity consistently led the frontal ERP were better correlated with the frontal ERP, compared to those that trailed the frontal ERP. (**C** and **D**) Correlation between the BF-ERP delay against the BF bursting index (**C**) and baseline firing rate (**D**). BF bursting neurons whose activity consistently led the frontal ERP showed stronger bursting responses (**C**) compared to those that trailed the frontal ERP, while having similar baseline firing rates (**D**).

The following figure supplements are available for figure 3:

**Figure supplement 1**. Additional analysis on the fine temporal relationship between BF bursting and the frontal ERP.

**Figure supplement 2**. A model of how BF bursting activity generates the frontal ERP.

followed the frontal ERP, the BF neurons that temporally led the frontal ERP were better correlated with the frontal ERP and showed stronger bursting responses. In other words, the contributions from BF bursting neurons that led or trailed the frontal ERP were not equal. These results support the hypothesis that the frontal ERP may be driven by the strongly bursting BF neurons that led the ERP by 5–10 msec through a fast circuit mechanism.

To further understand how the frontal ERP was generated, we recorded LFP activity across all cortical layers in frontal cortical regions below the EEG electrode. Concomitant with the frontal ERP and the BF bursting response, we observed clear LFP responses in a subset of cortical layers (*Figure 4A,B*, *Figure 4—figure supplement 1*, *Figure 4—figure supplement 2*). The non-uniform distribution of LFP responses across cortical layers indicates that these LFP responses were not volume conducted from distant sources.

Furthermore, we found that the layer profiles of LFP responses associated with BF bursting were highly similar between oddball and standard trials (*Figure 4A–C*, *Figure 4—figure supplement 1*, *Figure 4—figure supplement 2*). The amplitudes of LFP responses and BF bursting were significantly correlated in single trials (p<0.001) in 86% (66/77) of BF bursting neurons recorded simultaneously with frontal LFPs (*Figure 4D*). These results indicate that the LFP response amplitudes were linearly scaled between oddball and standard trials, and also linearly scaled with BF bursting amplitudes,

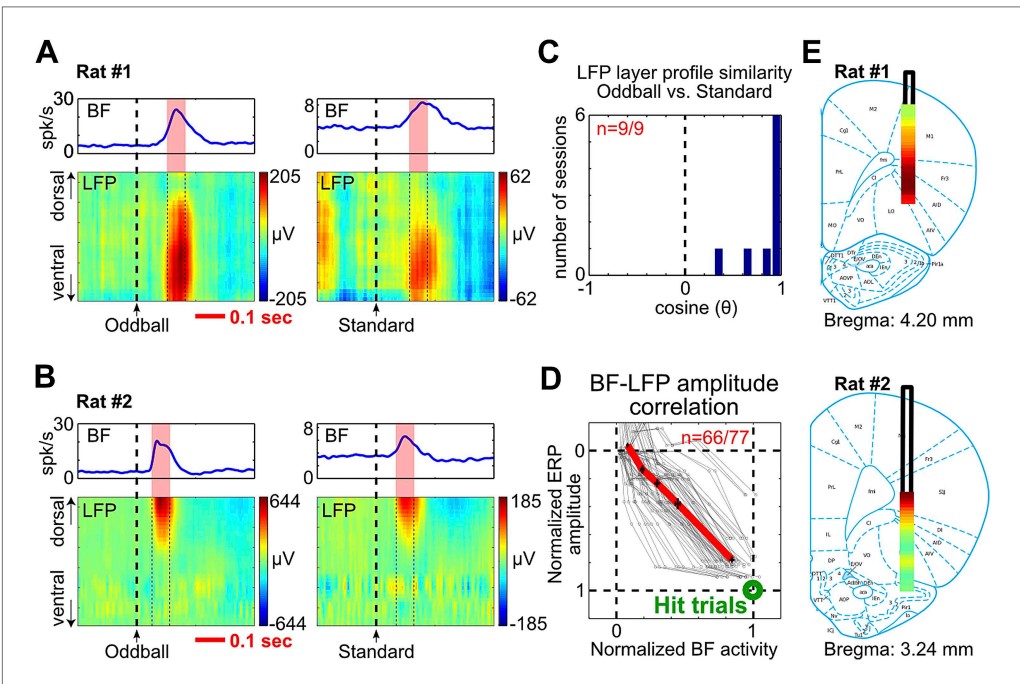

**Figure 4**. BF bursting is associated with LFP responses in deep layers of the frontal cortex. (**A** and **B**) LFP responses to tone onset across cortical layers in the frontal cortex in two representative rats. LFP responses occurred in the same window as BF bursting (red shaded area) despite variability in the latency of BF bursting in different animals and trial types. LFP responses were shown in different amplitude scales in oddball (left) and standard trials (right) to highlight the similar layer profile. (**C**) The layer profiles of LFP responses were similar (9/9 sessions) between oddball and standard trials. Similarity is defined as the cosine of the angle between the two 32-dimension LFP layer profile vectors. (**D**) 66/77 BF bursting neurons showed significant (p<0.001) single trial amplitude correlation and linear amplitude scaling with LFP responses. Conventions as in *Figure 2B*. (**E**) The LFP layer profiles overlaid on histological reconstructions show that the prominent positive LFP responses were located in the deep layers of the frontal cortex.

The following figure supplements are available for figure 4:

**Figure supplement 1**. LFP layer profiles in the oddball task are similar to those elicited by BF electrical stimulation.

**Figure supplement 2**. Additional examples.

similar to what we observed between BF bursting and the frontal ERP (*Figure 2B*). Therefore, the characteristic LFP layer profile associated with BF bursting response likely reflects the organization of local activity dipoles that generate the frontal ERP at the skull surface.

The most prominent feature of the LFP layer profile was a strong positive LFP response located within the deep cortical layers of the frontal cortex (*Figure 4E*, *Figure 4—figure supplement 1*, *Figure 4—figure supplement 2*), which are known to be the predominant target layers of cortical projections from the BF (*Henny and Jones, 2008*).

Finally, to further determine whether BF activity was sufficient to trigger the frontal ERP with a short delay, we investigated whether artificially inducing BF bursting activity can produce the characteristic LFP response in the oddball task. To reliably activate BF bursting neurons, electrical stimulation was chosen over more selective methods such as optogenetics because the neurochemical identity of BF bursting neurons needed for selective activation techniques remains to be established (*Lin and Nicolelis, 2008*).

We found that a single pulse of BF electrical stimulation in the absence of any auditory stimulus was able to elicit highly reliable LFP responses in frontal cortical regions, both under isoflurane anesthesia and during wakefulness (*Figure 5*, *Figure 4—figure supplement 1*, *Figure 4—figure supplement 2*). BF electrical stimulation generated positive LFP responses in the same deep cortical layers that showed strong positive LFP responses in the oddball task within 5–10 msec (*Figure 5*, *Figure 4—figure supplement 1*, *Figure 4—figure supplement 2*), thus recapitulating the signature features of oddball LFP responses—in terms of timing, layer profile and polarity of LFP responses. BF electrical stimulation also generated an earlier LFP response, possibly due to activation of other neuronal populations in the BF (*Gritti et al., 1997*, *2003*) or from antidromic activation of frontal cortex projections to the BF (*Zaborszky et al., 1997*). Together, these results support the idea that the BF bursting in the oddball task is sufficient to generate and account for key features of the frontal ERP response.

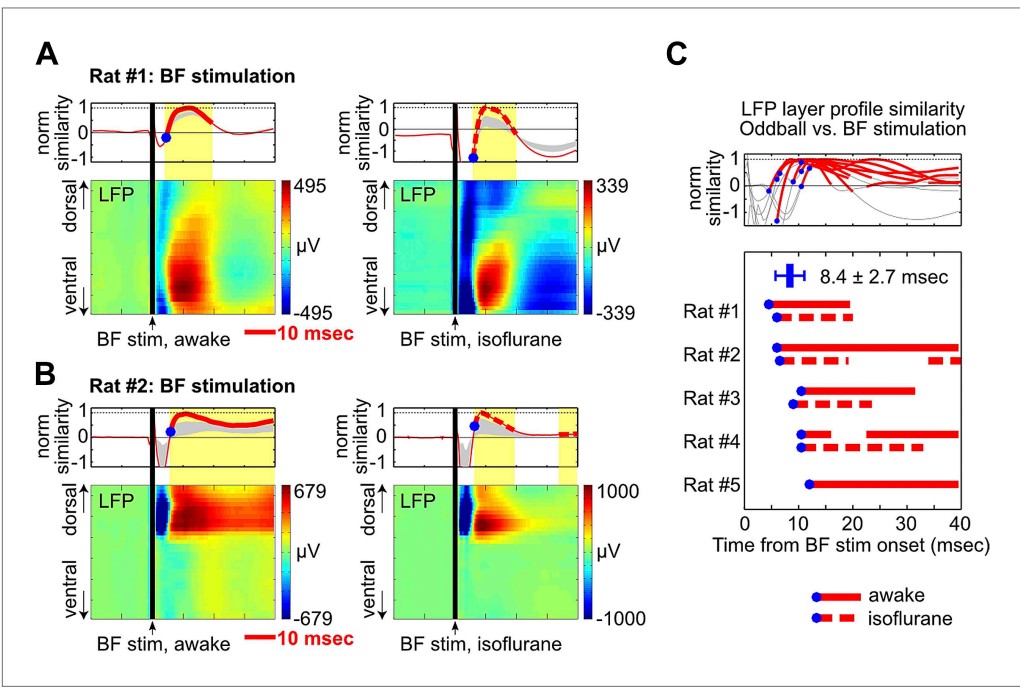

**Figure 5**. BF electrical stimulation mimics the layer profile of LFP responses associated with BF bursting. (**A** and **B**) LFP responses to a single pulse of BF electrical stimulation (70 µA per electrode), delivered either during wakefulness (left) or under isoflurane anesthesia (right), in the same two rats shown in *Figure 4A,B*. The top panels show the similarity between the LFP layer profile evoked by BF electrical stimulation and the LFP layer profile in the oddball task (red traces), defined as the scalar projection of the two layer profile vectors normalized by its peak amplitude. Gray shaded areas indicate the 95% confidence interval (permutation test). (**C**) Significantly similar layer profiles (thick red traces) started, on average, 8.4 ± 2.7 msec after BF electrical stimulation (blue circles). Each trace represents the similarity of layer profiles in one session. BF stimulation sessions during wakefulness and isoflurane are indicated by solid and dotted traces, respectively.

## Discussion

ERP is one of the oldest and also most widely used methods in neuroscience research (*Davis, 1939*; *Rohrbaugh et al., 1990*; *Luck, 2005a*; *Luck and Kappenman, 2012*). Contrary to the common belief that cognitive ERPs are generated by local activity within the cerebral cortex, we found that a frontal ERP response closely linked to behavioral performance in the auditory oddball task (*Figure 1*) is tightly coupled with BF bursting activity in single trials in terms of both amplitude (*Figure 2*) and timing (*Figure 3*). The coupling between BF bursting and the frontal ERP remains invariant in the face of different types of stimulus (oddball vs standard) and behavioral response (hit vs miss). The frontal ERP is likely generated by the underlying layer-specific LFP response, which is similarly coupled with BF bursting activity (*Figure 4*). BF electrical stimulation reproduces the key features of LFP responses in the oddball task (*Figure 5*), suggesting that the BF bursting activity is sufficient to generate and account for most of the frontal ERP response in the oddball task. The BF bursting activity therefore provides a novel neural correlate of the frontal ERP response. This is the first time, to the best of our knowledge, that a key component of cognitive ERPs has been linked to the activity of a well-defined neuronal population located outside of the cerebral cortex. These findings highlight the important and previously unrecognized role of long-range subcortical inputs from the BF in generating cognitive ERPs.

While the strong correlation between the frontal ERP and BF bursting activity does not rule out the possibility that the observed correlation might be generated by common inputs from other un-sampled structures, our finding that the activity of strongly bursting BF neurons temporally precedes the frontal ERP by 5–10 msec (*Figure 3*) is more compatible with BF driving the frontal ERP, and less compatible with the common input hypothesis. We suggest that even the weakly bursting BF neurons whose activity trails the frontal ERP may causally contribute to the frontal ERP with a short delay (*Figure 3—figure supplement 2*). Another possibility is that the subset of BF bursting neurons trailing the frontal ERP may be driven by inputs from the frontal cortex (*Zaborszky et al., 1997*). The hypothesis that BF bursting activity generates the frontal ERP is further supported by our finding that BF electrical stimulation triggers the same layer profile of oddball LFP responses within 5–10 msec (*Figure 5*), the same temporal delay found in the correlation analysis (*Figure 3*). Given the complex anatomical connection network associated with the BF region (*Semba, 2000*; *Zaborszky, 2002*; *Jones, 2003*), the goal of this study was not to fully dissect the causal contributions of various connection pathways. Rather, our goal was to test the specific hypothesis that the BF bursting activity alone, through the prominent projection to the frontal cortex, is sufficient to trigger and account for the oddball ERP response. While our results do not rule out the possible contribution of common inputs, our findings support the conclusion that the BF bursting activity alone is sufficient to trigger and account for most of the frontal ERP in the auditory oddball task.

Two key features of the coupling between BF bursting and the frontal ERP—the short temporal delay and the LFP layer profile concentrating in deep cortical layers—are consistent with the specific properties of BF projections to the frontal cortex. The average conduction latency of BF neurons to the frontal cortex, measured by antidromic stimulation, is less than 5 msec (*Aston-Jones et al., 1985*; *Reiner et al., 1987*). Moreover, the fast-conducting population of cortically-projecting BF neurons (latencies 1–4 msec) are most easily activated (with lowest activation threshold) when the stimulation electrode is placed in deep cortical layers of the frontal cortex (*Aston-Jones et al., 1985*; *Reiner et al., 1987*). Deep cortical layers in the frontal cortex (layer V–VI) are also the predominant target layers of cortical projections from the BF, which contain more than 70% of axonal varicosities from the BF (*Henny and Jones, 2008*). Among the BF axonal varicosities, the most abundant (~50%) is the large GABAergic terminal, while cholinergic terminals from the BF are smaller and fewer (15–20%) (*Freund and Meskenaite, 1992*; *Henny and Jones, 2008*). These specific properties of BF projection to the frontal cortex support our hypothesis that the frontal ERP is generated by BF inputs to deep cortical layers through a fast circuit mechanism.

A likely candidate to mediate the fast circuit mechanism is the non-cholinergic BF inputs because of their anatomical abundance (*Freund and Meskenaite, 1992*; *Gritti et al., 1997*; *Henny and Jones, 2008*) and their fast ionotropic actions. While studies of the BF have traditionally focused on its cholinergic neurons (*Everitt and Robbins, 1997*), the majority of BF corticopetal projections are non-cholinergic neurons, consisting mostly of GABAergic neurons and a smaller subset of glutamatergic neurons (*Gritti et al., 1997*; *Henny and Jones, 2008*). The GABAergic BF neurons in

particular preferentially innervate intracortical GABAergic interneurons and therefore may rapidly enhance cortical activity through disinhibition (*Freund and Meskenaite, 1992*; *Henny and Jones, 2008*). Consistent with such a suggestion, salience-encoding BF bursting neurons represent a physiologically homogeneous group of non-cholinergic BF neurons, which, unlike cholinergic BF neurons, do not change their mean firing rates between awake and sleep states (*Lin et al., 2006*; *Lin and Nicolelis, 2008*). The widespread spatial distribution of bursting neurons throughout multiple brain regions of the BF (*Figure 1—figure supplement 2*) is also consistent with the location of cortically-projecting BF neurons (*Aston-Jones et al., 1985*; *Reiner et al., 1987*; *Gritti et al., 1997*). Therefore, our results highlight the potential functional significance of the poorly understood non-cholinergic BF neurons (*Sarter and Bruno, 2002*; *Lau and Salzman, 2008*), and the importance in determining the neurochemical identity of salience-encoding BF bursting neurons. Future studies are needed to elucidate the precise nature of this ERP response, and test whether the frontal ERP reflects, as a result of BF modulation, the synchronous activation of pyramidal neuronal ensembles in the frontal cortex as commonly assumed (*Snyder, 1991*; *Miltner et al., 1994*; *Pascual-Marqui et al., 2002*; *Luck, 2005b*; *Nunez and Srinivasan, 2006*; *Riera et al., 2012*), or, alternatively, reflects the direct postsynaptic response of cortical neurons to non-cholinergic BF inputs.

The single trial coupling between BF bursting activity and the frontal ERP in the current study suggests that both neural signals are functionally homologous and reflect the motivational salience of attended stimuli (*Lin and Nicolelis, 2008*; *Avila and Lin, 2014*). In support of this view, behaviorally irrelevant stimuli presented passively to the animal elicit little response in both the BF neurons (*Lin and Nicolelis, 2008*) and the frontal ERP (*Figure 1C*). This observation indicates that both neural signals do not reflect passive sensory processing. Instead, the BF bursting activity, and likely the frontal ERP response, reflects motivational salience irrespective of sensory modality or stimulus identity (*Lin and Nicolelis, 2008*; *Avila and Lin, 2014*). This motivational salience signal is highly relevant for behavioral performance, which predicts successful detection of a near-threshold sound (*Lin and Nicolelis, 2008*) as well as performance in the oddball task (*Figure 1E*). It is important to point out that the relationship between BF activity and behavioral performance is more complex and involves more than the phasic bursting response that encodes motivational salience. In behavioral contexts that require inhibition, such as Nogo trials in the Go/Nogo task, BF neurons display an initial bursting response followed by a subsequent sustained inhibition, with the latter response better correlated with behavior (*Lin and Nicolelis, 2008*). Future studies will need to determine the contribution of BF inhibitory responses on behavioral performance when animals need to stop or cancel a behavioral response instead of making a response to motivationally salient stimuli. It will also be important to determine whether the BF inhibitory response leads to a corresponding frontal ERP signature.

Based on these observations, we propose that the main function of non-cholinergic BF neurons is to serve as a signal amplifying mechanism for motivationally salient stimuli. This amplification mechanism is needed because most sensory stimuli animals and humans encounter are behaviorally irrelevant and not processed further by the brain. For the subset of stimuli that are motivationally salient and predict important outcomes, the brain therefore requires an additional system to amplify its processing based on the motivational, but not perceptual, salience of the stimulus. The non-cholinergic BF neurons represent an ideal candidate for this amplification mechanism, which not only needs to encode the motivational salience of the attended stimulus but also needs to powerfully modulate and amplify cortical activity as fast as possible.

The current study supports this hypothesis by demonstrating that the motivational salience information encoded by BF bursting activity is transformed into the frontal ERP response with a minimum delay, which likely provides powerful modulation and amplification on the early stages of information processing. Future studies need to determine precisely how task-related single neuron activity in the frontal cortex is modulated by BF inputs and the resulting frontal ERP. Ultimately, this proposed amplification mechanism should lead to faster and better decision making. In support of this prediction, we recently showed that stronger BF motivational salience signal is quantitatively coupled with faster and more precise decision speed (*Avila and Lin, 2014*).

The findings in this study likely can be extended to primates because BF neurons with bursting responses to motivationally salient stimuli are similarly found in monkeys (*DeLong, 1971*; *Wilson and Rolls, 1990*; *Richardson and Delong, 1991*), and BF electrical stimulation in monkeys similarly triggers LFP responses in the frontal cortex (*Richardson and Fetz, 2012*). Despite the many differences in rodent and human ERP studies (such as species, the use of primary reward and punishment in rodents

but not in humans, the choice of electrical reference, etc), the major ERP components are broadly similar in rodents and humans (*Yamaguchi et al., 1993*; *Ehlers et al., 1994*; *Shinba, 1997*; *Sambeth et al., 2003*). The rodent frontal ERP described in this study is most similar to the processing negativity or Nd ERP component in humans (*Hansen and Hillyard, 1980*; *Näätänen, 1982*; *Näätänen et al., 1987*) because both the rodent frontal ERP and the human processing negativity responses share several features—negative polarity of the ERP, early timing in the N1 ERP window, spatial distribution centered on the frontal cortex, as well as reflecting motivational salience or selective attention toward the stimulus. Further studies are needed to establish the functional homology between the rodent and human frontal ERP components, and determine the potential role of non-cholinergic BF neurons in selective attention in humans.

An important implication of this study is that rodent ERPs may serve as a unique translational platform to bridge the human ERP literature and the neurophysiology literature in animal models. Studies in rodent models may provide unique mechanistic insights that can inform human ERP studies. For example, localizing the sources of human ERP responses, that is solving the 'inverse problem', requires assumptions about the number and spatial distribution of underlying sources (*Pascual-Marqui et al., 2002*; *Luck, 2005b*). Our results suggest that, at least in the case of the frontal ERP component, the underlying source may reflect the anatomical projection pattern of the salience-encoding BF neurons, which project extensively to broad regions of the cerebral cortex (*Gritti et al., 1997*; *Zaborszky, 2002*; *Henny and Jones, 2008*; *Chandler et al., 2013*). Our results also raise the novel hypothesis that the decline of frontal ERP amplitude and associated cognitive functions in conditions such as schizophrenia, ADHD, Alzheimer's disease and cognitive aging (*Schreiber et al., 1992*; *Iragui et al., 1993*; *Polich, 1997*; *Barry et al., 2003*; *Caravaglios et al., 2008*) may result from the functional impairment of the poorly understood non-cholinergic basal forebrain system or a dysfunctional BF-cortical interaction. These results also suggest that the increasing use of deep brain stimulation of the BF to ameliorate dementia (*Hescham et al., 2013*; *Salma et al., 2014*) may be mediated in part by activating non-cholinergic BF neurons and perhaps compensating for their functional impairment in these conditions.

## Materials and methods

### Subjects

Eight male Long Evans rats (Charles River, NC), aged 3–6 months and weighing 300–400 grams at the start of the experiment, were used for this experiment. Rats were housed under 12/12 day/night cycle with *ad libitum* access to rodent chow and water in environmentally controlled conditions. During training and recording procedures, rats were mildly water restricted to their 90% weight and were trained in a daily session of 60–90 min in length, five days a week. Rats received 15 min water access at the end of each training day with free access on weekends. All experimental procedures were conducted in accordance with the National Institutes of Health (NIH) Guide for Care and Use of Laboratory Animals and approved by the National Institute on Aging Animal Care and Use Committee.

### Apparatus

12 plexiglass operant chambers (11'L × 8 ¼ 'W × 13'H), custom-built by Med Associates Inc (St. Albans, VT) were contained in sound-attenuating cubicles (ENV-018MD) each with an exhaust fan that helped mask external noise. Each chamber was equipped with one photo-beam lickometer reward port (CT-ENV-251L-P) located in the center of the front panel, with its sipper tube 7.5 cm above the grid floor. Two infrared (IR) sensors were positioned to detect reward port entry and sipper tube licking, respectively. Water reward was delivered through a custom-built multi-barrel sipper tube. The delivery system was controlled by pressurized air (2.6 psi) and each solenoid opening (10 msec) was calibrated to deliver a 10 µl drop of water. The reward port was flanked by two nosepoke ports (ENV-114M), located 6.6 cm to each side and 5.9 cm above the grid floor. The nosepoke ports were inactive in the behavioral tasks.

Each chamber was equipped with two ceiling-mounted speakers (ENV-224BM) to deliver auditory stimuli controlled by an audio generator (ANL-926), and two stimulus lights (ENV-221) positioned above the reward port in the front panel. Behavior training protocols were controlled by Med-PC software (Version IV), which stored all event timestamps at 2 msec resolution and also sent out TTL signals to neurophysiology recording systems to register event timestamps.

## Behavioral training

Rats were initially trained in a simple tone-reward association task, in which a 6 kHz tone (0.5 s long, 70 dB) signaled the availability of reward if responded within 3 s. Rats were rewarded with 3–5 drops of water reward starting at the 3$^{rd}$ lick of the sipper tube. The inter-trial intervals (ITI) were randomly drawn from 5, 8, 11, or 14 s. If the rat licked outside the 3 s reward window (false alarm), the ITI timer was reset. After completing more than 180 trials in a 60 min session, rats were transitioned to the auditory oddball task.

The auditory oddball task contains infrequent-rewarded (oddball, 6 kHz) as well as frequent-unrewarded (standard, 10 kHz) tones, delivered through the same speaker. The time between stimulus presentations was 2 s, and the number of standards that were presented in between oddball tones was uniformly drawn from 2, 3, 4, 5, and 6, corresponding to 6–14 s ITI between oddball tones. When the oddball tone was presented, the rules for receiving reward were the same as that for the tone-reward association task. False alarms during the oddball task reset the ITI timer. Correct behavioral response to the oddball tone (hit) led to reward delivery, as well as a temporary cessation of any tone presentation and the ITI timer until the end of reward consumption. In the passive oddball task (*Figure 1C*), the same stimuli were presented to rats that were not water deprived, and the access to reward port was physically blocked to prevent responding.

## Stereotaxic surgery and electrode

After reaching asymptotic behavioral performance, rats were taken off water restriction for at least 3 days before undergoing stereotaxic surgery for chronic electrode implant. Rats were anesthetized with isoflurane (4% isoflurane induction followed with 1–2% maintenance) and received atropine (0.02–0.05 mg/kg, i.m.) to reduce respiratory secretion. Ophthalmic ointment was applied to prevent corneal dehydration. A heating pad was used to maintain body temperature at 37°C. Rats were placed in a stereotaxic frame (David Kopf Instrument, CA) fitted with atraumatic earbars.

After the skull was exposed, up to three types of probes were chronically implanted in one animal: (1) EEG skull screws were implanted in contact with the dura over the frontal cortex (AP 3.0 mm, ML 3.0 mm relative to Bregma [*Paxinos and Watson, 2007*]) and the primary visual cortex (AP–7.0 mm, ML 4.5 mm); (2) a NeuroNexus linear probe (A1-style, 100 µm spacing, 32-channel) was slowly lowered into the frontal cortex (AP 3.0–4.0 mm, ML–3.0 mm) with the target depth at 4–6.5 mm below cortical surface; (3) A custom-built 32-wire multi-electrode moveable bundle was implanted into bilateral BF (AP–0.5 mm, ML +/− 2.25 mm, DV 7 mm below cortical surface). The electrode consisted of two moveable bundles, each containing 16 polyimide-insulated tungsten wires (California Fine Wire) ensheathed in a 28-gauge stainless steel cannula and controlled by a precision microdrive. Eight of the wires in a bundle were 38 µm in diameter and the other eight were 16 µm diameter, with 0.1–0.3 MΩ impedance measured at 1 kHz (niPOD, NeuroNexusTech, MI). During surgery, the cannulae were lowered to DV 6–6.3 mm below cortical surface using a micropositioner (Model 2662, David Kopf Instrument), and the electrodes were advanced to 7 mm below cortical surface. Two of the 32 channels were replaced by silver wire pigtails that were connected with skull screws during surgery to record EEG signals. A common ground screw and a separate reference screw were placed over the right cerebellum (AP–10 mm, 3.0 mm) and left cerebellum (AP–10 mm, −3.0 mm), respectively. The electrode and screws were covered with dental cement (Hygenic Denture Resin).

Rats received acetaminophen (300 mg/kg, oral) and topical antibiotics after surgery for pain relief and prevention of infection. Water restriction and behavioral training resumed 7–10 days after surgery. Cannulae and electrode tip locations were verified with cresyl violet staining of histological sections at the end of the experiment. The tips of the linear probes in the frontal cortex are indicated by arrows in *Figure 4—figure supplement 1* and *Figure 4—figure supplement 2*. BF electrodes were located between −0.12 mm–−0.84 mm relative to Bregma, as shown in *Figure 1—figure supplement 2A*.

## Recording

Electrical signals were referenced to a common skull screw placed over the cerebellum. Electrical signals were filtered (0.03 Hz—7.5 kHz) and amplified using Brighton Omnetics or Cereplex M digital headstages and recorded using a Neural Signal Processor (Blackrock Microsystems, UT). EEG and LFP signals were continuously digitized and saved to disk at a rate of 1 kHz or 2 kHz (band-pass filtering between 0.7–500 Hz). In general, 32 channels were devoted to laminar (Neuronexus probe) LFP recordings, 2 channels for skull EEG recording, and 30 channels for LFP & single unit recording in the BF region.

Spike waveforms were further filtered (250 Hz–5 kHz) and digitized at 30 kHz and stored to disk only when the waveform exceeded a user-defined amplitude threshold. Spike waveforms were sorted offline using OfflineSorter (Plexon Inc, TX). Only single units with clear separation from the noise cluster and with minimal (<0.1%) spike collisions were used for further analyses. Additional cross correlation analysis was used to remove duplicate units recorded simultaneously over multiple electrodes. When multiple sessions from the same animal were used, BF electrodes were advanced at least 125 µm in between sessions to sample from distinct BF single neuron ensembles. The linear probe in the frontal cortex remained fixed across sessions. A total of 168 BF neurons was recorded simultaneously with EEG or LFP activity in the frontal cortex. *Table 1* provides detailed information about the types of neural signals recorded in each animal, and how the data were used for different analyses in each figure.

## Neuronal discrimination of behavioral performance

Signal detection theory (*Green and Swets, 1966*) was used to quantify neuronal discrimination of successful (hit) vs failed (miss) tone detections. Receiver Operating Characteristic (ROC) curves, in particular the area under curve (AUC) measure, was used to determine how neuronal responses in hit trials and miss trials differed. The non-parametric AUC measure, referred to as *choice probability* in *Figure 1E* and *Figure 1—figure supplement 1*, quantified the difference of the average EEG activity within a 100 msec window between hit trials vs miss trials (or between oddball vs standard trials). A choice probability of 0.5 represents a complete overlap between the two distributions, while a choice probability of 1 (or 0) indicates complete non-overlap between the two distributions, with one distribution having larger (or smaller) values. The statistical significance of the choice probability was established by comparing against a null distribution of choice probability values generated by randomly permuting the identity of hit and miss trials 1000 times (p<0.01, two-sided). The choice probability (and its statistical significance) was generated for each EEG signal, at time lags between [−0.2, 0.4] sec of tone onsets with 10 msec steps. Therefore, a significant choice probability >0.5 indicated that the EEG signal at that time lag (within a 100 msec window) was significantly more positive for hit trials compared to miss trials. Conversely, a significant choice probability <0.5 indicated that the EEG signal at that time lag was significantly more negative for hit trials compared to miss trials.

## Identification of BF bursting neurons

To identify BF bursting neurons, we calculated a bursting index for each BF neuron, defined as the ratio of the bursting amplitude (average firing rate in the [50, 200] msec window in hit trials) over the average firing rate of the neuron in the entire session. Based on the peri-stimulus time histograms (PSTHs, *Figure 1—figure supplement 2B*), bursting index of 2.5 was chosen as the cutoff for classifying BF bursting neurons, which accounted for 57% (96/168) BF neurons. BF bursting neurons have highly homogeneous baseline firing rates (3.77 ± 1.48 spikes/sec, mean ± std, maximum 6.36 spikes/sec), while other BF neurons have heterogeneous baseline firing rates (5.34 ± 6.25 spikes/sec, mean ± std, maximum 31.4 spikes/sec).

## BF-ERP and BF-LFP amplitude correlation

To determine the single trial amplitude correlation between the activity of individual BF neurons and the amplitude of EEG/LFP signals, we calculated the number of spikes for each BF neuron and the average amplitude of EEG/LFP activity in the [50, 200] msec window following the onset of all oddball and standard tones. The statistical significance of BF-ERP amplitude correlation was determined using Pearson correlation (p<0.001). R- and p-values for all BF neurons are shown in *Figure 2D*.

For the purpose of visualizing the linear amplitude scaling relationship between BF bursting and EEG/LFP (*Figure 2B,C and 4D*), all trials were first binned into five quintiles based on the population bursting amplitude of all BF bursting neurons recorded simultaneously in a session. The mean amplitude of BF bursting and EEG/LFP in the five quintiles were then normalized by their respective average amplitudes in hit trials, such that the average hit response amplitude corresponded to 100%.

## Analysis of BF-ERP fine temporal relationship

To determine the fine temporal relationship between BF bursting response and the frontal ERP at msec temporal resolution (*Figure 3*, *Figure 3—figure supplement 1*), we calculated cross correlations between the activity of individual BF bursting neurons and the frontal EEG signal at the [0, 170] msec window relative to tone onsets. To focus the analysis on BF bursting, the [0, 170] msec window was chosen to center on the BF bursting response. Only trials (both oddball and standard) with BF population

bursting amplitude greater than 30% of the average hit trial bursting amplitude were included in this analysis.

Normalized cross correlation (correlation coefficient) was calculated between two concatenated vectors corresponding to BF and EEG activity (described next), between lags [−50, 50] msec at 1 msec resolution. Specifically, for each BF bursting neuron, the spike train was binned at 1 msec, and the binned activity in the [0, 170] msec window from selected trials were concatenated into a long vector, each trial flanked by a 100 msec segment set to a fixed value corresponding to the mean firing rate of this neuron. For the EEG signal, EEG was first down sampled to 1 kHz sampling rate (1 msec bin), and the activity in the [0, 170] msec window from selected trials were similarly concatenated in a long vector, each trial flanked by a 100 msec segment set to a fixed value corresponding to the mean of the EEG signal. The 100 msec fixed value flanking each trial ensured that the activity in different trials remains segregated in the cross-correlation analysis. Statistical significance of the correlation coefficient was established by comparing against a null distribution of correlation coefficients generated by randomly permuting the trial order of EEG activity 1000 times ($p < 0.01$, two-sided).

## LFP layer profile and similarity

To determine the layer profile of LFP responses in association with BF bursting, we identified a 60 msec window around the peak of BF bursting response in oddball and standard trials, indicated by the red shaded areas in *Figure 4A,B* and *Figure 4—figure supplement 1*. The LFP layer profile was defined as the mean LFP response in this 60 msec window, represented by a 32-dimension vector. The similarity index (*Figure 4C*) between LFP layer profiles in oddball and standard trials was defined as the cosine of the angle between the two 32-dimension vectors. This similarity index measures how well the two vectors are aligned with each other irrespective of amplitude. The maximum value of this similarity index is 1 when the angle is 0°, and the minimum value is −1 when the angle is 180°. Statistical significance of the similarity index was established by comparing against a null distribution of similarity index values generated by randomly permuting the 32 dimensions of the LFP layer profile vector 1000 times ($p < 0.01$, two-sided). Since the LFP layer profiles in oddball and standard trials were statistically similar in all sessions (9/9) (*Figure 4C*), the overall LFP layer profile for each session was generated by averaging all trials, including both oddball and standard trials. The amplitudes of the overall LFP layer profile were normalized to its peak positive value among all cortical layers. As a result, the prominent positive LFP responses restricted in deep cortical layers of the frontal cortex have values of up to 1 (*Figure 4E*, *Figure 4—figure supplement 1*, *Figure 4—figure supplement 2*).

## BF electrical stimulation

To assess whether BF activity was sufficient to generate the frontal ERP response, we used BF electrical stimulation to mimic BF ensemble bursting activity with the goal of recreating the major features of the layer-specific LFP responses in the oddball task. Five rats were used for BF electrical stimulation over nine sessions. BF electrical stimulation was delivered while rats were awake but not performing any behavioral task (n = 5) or under isoflurane anesthesia (n = 4) (*Table 1*).

BF electrical stimulation was conducted in separate sessions after BF single unit activity in the oddball task has been recorded. BF stimulation was delivered through all BF electrodes used in the recording experiment. This was intended to mimic the widespread presence of BF bursting neurons throughout the recording region, representing an ensemble-bursting event of the entire population (*Lin et al., 2006*; *Lin and Nicolelis, 2008*; *Avila and Lin, 2014*). Furthermore, individual BF neurons tend to project to multiple subregions in the frontal cortex, unlike single neurons in other neuromodulatory systems which tend to project to one single subregion in the frontal cortex (*Chandler et al., 2013*). This result suggests that the activation of any subset of cortically-projecting BF neurons by electrical stimulation should provide similar modulation of the entire frontal cortex, irrespective of the exact location of BF electrical stimulation electrodes within this region.

One pulse of stimulation was delivered every 2 s, without any accompanying auditory stimuli. Individual stimulation pulse was a biphasic charge-balanced pulse (0.1 ms each phase) delivered through a constant current stimulator (stimulus isolator A365R, World Precision Instruments, FL), driven by a Master-8-VP stimulator (AMPI, Israel). Currents were flowing through all BF electrodes in one hemisphere against all BF electrodes in the other hemisphere, and never through EEG skull screws, or through ground or reference skull screws. Stimulation current level was set at 62–71 µA per electrode, resulting in a total of 1 mA over all electrodes. LFP signals from the frontal cortex were recorded using

a unit gain analog headstage (Blackrock Microsystem), sampled at 2 kHz with 0.7–500 Hz filter. When BF electrical stimulation was delivered under isoflurane anesthesia, rats were first anesthetized with 3% isoflurane for at least 5 min before the start of the experiment, and maintained at 2% isoflurane under a nose cone. The body of the rat was wrapped in an aluminum foil shield lined with paper towels such that the body did not make contact with any metal.

To assess the similarity of the LFP responses elicited by BF stimulation with the LFP layer profile in the oddball task, we projected the 32-dimension LFP layer vector elicited by BF stimulation onto the 32-dimension LFP layer profile in the oddball task at each time point, and calculated the length of the vector projection (scalar projection). The amplitude of the scalar projection was then normalized by its peak amplitude (excluding the 5 msec around electrical stimulation to avoid stimulation artifact), which we referred to as normalized similarity index (*Figure 5*, *Figure 4—figure supplement 1*, *Figure 4—figure supplement 2*). The normalized similarity index takes into account both the angle between the two LFP layer vectors and also the amplitude of the LFP responses elicited by BF electrical stimulation. Statistical significance of the normalized similarity index was established by comparing against a null distribution of scalar projections generated by randomly permuting the 32 dimensions of the LFP layer profile vector 1000 times ($p<0.01$, two-sided).

## Acknowledgements

This research is funded by the intramural research program of the National Institute on Aging, NIH and by NARSAD Young Investigator Award to SL. We thank JD Mayse, HMV Manzur, A Scaglione, I Avila, R Greenfield, J Long and PR Rapp for critical discussions and reading of the manuscript; G Nelson, I Avila and B Brock for technical help.

## Additional information

### Funding

| Funder | Grant reference number | Author |
|---|---|---|
| National Institutes of Health | | David P Nguyen, Shih-Chieh Lin |
| Brain & Behavior Research Foundation (NARSAD) | Young Investigator Award | Shih-Chieh Lin |

The funders had no role in study design, data collection and interpretation, or the decision to submit the work for publication.

### Author contributions

DPN, Designed the experiment jointly with S-CL, Performed experiments, Collected data, Preprocessed data; S-CL, Conceived the study, Designed the experiment jointly with DN, Interpreted data, Performed statistical analysis, Wrote the manuscript

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
