## [Decision Letter]

Thank you for sending your work entitled “A frontal cortex event-related potential driven by the basal forebrain” for consideration at *eLife*. Your article has been favorably evaluated by a Senior editor, a member of the BRE, and 2 reviewers.

The following individuals responsible for the peer review of your submission have agreed to reveal their identity: Howard Eichenbaum (Reviewing editor); Geoffrey Schoenbaum and Ben Hayden (peer reviewers).

The Reviewing editor and the reviewers discussed their comments before we reached this decision, and the Reviewing editor has assembled the following comments to help you prepare a revised submission.

In this study Nguyen and Lin use an auditory oddball task to study a new question: what is the source of the ERP signal? While conventional theories hold that it likely derives from local cortical processing, there's not a lot of strong evidence in favor of this hypothesis. Nguyen and Lin favor an alternative: the ERP depends on inputs from the basal forebrain bundle, a structure whose responses have previously been shown to encode salience. The signals are temporally and functionally linked to non-cholinergic BF neurons that encode salience.

In the current study, the authors report correlations between BF neuron firing and frontal ERP and LFP's recorded in rats performing an oddball discrimination task. Rats had to discriminate an oddball tone from an ongoing sequence in order to know when to lick to receive a fluid reward. The authors show that successful responding to the oddball tone was associated with bursting in BF neurons and with a frontal ERP. Further they show that the BF bursting was correlated with and led the frontal ERP and subsequent LFP recordings. Further, electrical activation of BF in both awake and anesthetized rats was sufficient to reproduce a similar LFP response. The authors argue that these data challenge the assumption that ERP measures reflect local processing.

The reviewers concluded that this is a strong paper, well supported and well argued. The experiments are convincing and well done. At the same time, the reviewers had some recommendations for improvement that should be addressed:

*Reviewer 1*:

My general issue with the interpretation is that it is unclear to me that the results necessarily challenge at least a weak version of the hypothesis that ERP's and LFP's reflect local processing or functions. In other words, while they certainly show that these measures are influenced by or can be affected by large-scale afferent input, it seems to me that this is not necessary a shocking result. In fact to some extent this would be as expected I think – that any manipulation that altered the frontal networks in significant ways might impact these measures, even if they reflect local processing. To the extent that this is the first demonstration of such a thing, I think it is significant. And as the authors point out, this has implications for how diseases affecting BF and subcortical areas might impact prefrontal processing. But I think the black and white assertion that the authors make is unnecessary and a bit of a straw man. It is not a surprise that subcortical systems affect processing in the cortical mantle. In fact this is what such inputs are supposed to do – the real question – and perhaps a logical next step to suggest is how they affect the processing. What local information processing or function is specifically lost when BF does not burst?

To assess this, it would be necessary to do two things that the authors did not do. The first would be to eliminate the bursting instead of inducing it. Generally I think this is more effective since it shows that the function requires the input. Though I think it is unlikely here, it is possible that activation is causing something that does not normally occur. And the second would be to relate the manipulation to behavior. Even in the context of the current report, this would have been terrific – does artificial activation of the BF neurons facilitate correct hits? Or does inhibition cause misses?

Lastly I wish the authors had reversed their task. As it stands, I am somewhat concerned that the electrical activity is only observed when the animal makes a motor response – licking. I assume the authors think it reflects a function – recognizing the oddball tone and redirecting behavior generally. But what is the evidence of this? Would it be the same if the rat had to stop licking at the oddball tone? Some discussion of this would be helpful.

No further experiments are needed here, but the results should be framed in a way to acknowledge these issues.

*Reviewer 2*:

I only had one major comment and this is with how the paper is framed. The authors describe two competing hypotheses, that ERP is generated by local cortical processing and that it is generated by BF neurons. But I don't think these hypotheses are in conflict. It seems that ERP may be derived from local processing that is in turn driven by BF firing. Indeed, the authors seem to acknowledge this when they say that “our results support the alternative hypothesis that some ERP components reflect how CC is modulated by subcortical inputs. ” Which seems like an acknowledgement that both may be right. It seems that their data cannot answer the question they ask: do ERPs come from BF or from CC? Instead they answer a different (and also very important) question, what is it that ultimately drives the ERP? And the answer is “salience signals in BF” although they may do so through changing cortical targets. I still think these results are nonetheless important; I just think they should be framed differently.

---

## [Author Response]

Reviewer 1:

*My general issue with the interpretation is that it is unclear to me that the results necessarily challenge at least a weak version of the hypothesis that ERP's and LFP's reflect local processing or functions. In other words, while they certainly show that these measures are influenced by or can be affected by large-scale afferent input, it seems to me that this is not necessary a shocking result. In fact to some extent this would be as expected I think* – *that any manipulation that altered the frontal networks in significant ways might impact these measures, even if they reflect local processing. To the extent that this is the first demonstration of such a thing, I think it is significant. And as the authors point out, this has implications for how diseases affecting BF and subcortical areas might impact prefrontal processing. But I think the black and white assertion that the authors make is unnecessary and a bit of a straw man. It is not a surprise that subcortical systems affect processing in the cortical mantle. In fact this is what such inputs are supposed to do* – *the real question* – *and perhaps a logical next step to suggest is how they affect the processing. What local information processing or function is specifically lost when BF does not burst*?

We agree with the reviewer’s comment and have reframed our question accordingly. The study now seeks to determine whether subcortical inputs play a significant role in generating cognitive ERPs, as a complementary hypothesis to the conventional view that ERPs are generated by local cortical activity alone.

We also point out that the precise nature of the frontal ERP response remains to be determined in future studies and there are two possible scenarios: The frontal ERP may reflect, as a result of BF modulation, the synchronous activation of pyramidal neuronal ensembles in the frontal cortex as commonly assumed. Alternatively, the frontal ERP may reflect the direct postsynaptic response of cortical neurons to non-cholinergic BF inputs.

To better understand how cortical processing is affected by BF inputs, it will be important in future studies to determine precisely how task-related single neuron activity in the frontal cortex is modulated by BF inputs and the resulting frontal ERP, under behavioral task conditions as well as under anesthesia. The current study was not able to address this issue because the multi-site linear probe was not able to record single unit activity in the frontal cortex after it had been chronically implanted and remained in the same position after two weeks post-implant.

*To assess this, it would be necessary to do two things that the authors did not do. The first would be to eliminate the bursting instead of inducing it. Generally I think this is more effective since it shows that the function requires the input. Though I think it is unlikely here, it is possible that activation is causing something that does not normally occur*.

We agree with the reviewer that selective inhibition of BF bursting under behavioral contexts will provide a stronger support of our hypothesis. A suitable technique to achieve this goal is optogenetics, which provides the required spatial and temporal resolution to transiently inhibit BF neurons in behavioral contexts. Because of the anatomical heterogeneity in the BF region that contains three major corticopetal projection populations, we had chosen to develop optogenetic approaches in a separate study using transgenic mice instead of rats. Using transgenic mice will allow us to determine the neurochemical identity of salience-encoding BF neurons, as well as providing the necessary tools to selectively manipulate the activity in subsets of BF neurons.

*And the second would be to relate the manipulation to behavior. Even in the context of the current report, this would have been terrific* – *does artificial activation of the BF neurons facilitate correct hits? Or does inhibition cause misses*?

While the BF motivational salience signal is highly relevant for behavioral performance, which predicts successful detection of a near-threshold sound (24) as well as performance in the oddball task (Figure 1), the relationship between BF activity and behavioral performance is more complex and involves more than the phasic bursting response that encodes motivational salience. For example, in behavioral contexts that require inhibition, such as Nogo trials in the Go/Nogo task, BF neurons display an initial bursting response followed by a subsequent sustained inhibition, with the latter response better correlated with behavior (24).

Because of the complex dynamics of both bursting and inhibitory responses in the BF, we did not attempt to manipulate the behavioral performance in the current study using the non-selective electrical stimulation technique. However, in a related study that does not involve behavioral inhibition, BF electrical stimulation that augments the strength of BF bursting leads to faster and more precise reaction time (2). We will revisit this issue of manipulating BF activity in behavioral contexts in future studies using optogenetic approaches that can bidirectionally manipulate the activity of selected BF neurons.

*Lastly I wish the authors had reversed their task. As it stands, I am somewhat concerned that the electrical activity is only observed when the animal makes a motor response* – *licking. I assume the authors think it reflects a function* – *recognizing the oddball tone and redirecting behavior generally. But what is the evidence of this? Would it be the same if the rat had to stop licking at the oddball tone? Some discussion of this would be helpful*.

We agree with the reviewer that this is an important and unresolved question, especially considering the potential role of BF inhibitory responses and its association with Nogo responses discussed earlier. Future studies will need to determine the contribution of BF inhibitory responses on behavioral performance when animals need to stop or cancel a behavioral response instead of making a response to motivationally salient stimuli. It will also be important to determine whether the BF inhibitory response leads to a corresponding frontal ERP signature.

Reviewer 2:

*I only had one major comment and this is with how the paper is framed. The authors describe two competing hypotheses, that ERP is generated by local cortical processing and that it is generated by BF neurons. But I don't think these hypotheses are in conflict. It seems that ERP may be derived from local processing that is in turn driven by BF firing. Indeed, the authors seem to acknowledge this when they say that “our results support the alternative hypothesis that some ERP components reflect how CC is modulated by subcortical inputs. ” Which seems like an acknowledgement that both may be right. It seems that their data cannot answer the question they ask: do ERPs come from BF or from CC? Instead they answer a different (and also very important) question, what is it that ultimately drives the ERP? And the answer is “salience signals in BF” although they may do so through changing cortical targets. I still think these results are nonetheless important; I just think they should be framed differently*.

We agree with the reviewer’s comment and have reframed our question accordingly. The study now seeks to determine whether subcortical inputs play a significant role in generating cognitive ERPs, as a complementary hypothesis to the conventional view that ERPs are generated by local cortical activity alone.

We also point out that the precise nature of the frontal ERP response remains to be determined in future studies and there are two possible scenarios: The frontal ERP may reflect, as a result of BF modulation, the synchronous activation of pyramidal neuronal ensembles in the frontal cortex as commonly assumed. Alternatively, the frontal ERP may reflect the direct postsynaptic response of cortical neurons to non-cholinergic BF inputs.